# Societal decisions about climate mitigation will have dramatic impacts on eutrophication in the 21st century

E. Sinha [1,2,5], A.M. Michalak [1,2], K.V. Calvin [3] & P.J. Lawrence [4]

Excessive nitrogen runoff leads to degraded water quality, harming human and ecosystem health. We examine the impact of changes in land use and land management for six combinations of socioeconomic pathways and climate outcomes, and find that societal choices will substantially impact riverine total nitrogen loading (+54% to −7%) for the continental United States by the end of the century. Regional impacts will be even larger. Increased loading is possible for both high emission and low emission pathways, due to increased food and biofuel demand, respectively. Some pathways, however, suggest that limiting climate change and eutrophication can be achieved concurrently. Precipitation changes will further exacerbate loading, resulting in a net increase of 1 to 68%. Globally, increases in cropland area and agricultural intensification will likely impact vast portions of Asia. Societal and climate trends must therefore both be considered in designing strategies for managing inland and coastal water quality.

[1] Department of Global Ecology, Carnegie Institution for Science, 260 Panama St., Stanford, CA 94305, USA. [2] Department of Earth System Science, Stanford University, 473 Via Ortega, Room 140, Stanford, CA 94305, USA. [3] Joint Global Change Research Institute, Pacific Northwest National Laboratory, 5825 University Research Court, Suite 3500, College Park, MD 20740, USA. [4] Earth System Laboratory, National Center for Atmospheric Research, 1850 Table Mesa Drive, Boulder, CO 80305, USA. [5] Present address: Atmospheric Sciences & Global Change Division, Pacific Northwest National Laboratory, 902 Battelle Boulevard, Richland, WA 99352, USA. Correspondence and requests for materials should be addressed to E.S. (email: esinha@stanford.edu) or to A.M.M. (email: michalak@stanford.edu)

Excessive nitrogen runoff is the primary cause of eutrophication in estuaries and coastal waters[1]. This eutrophication has led to increases in the occurrence, frequency, and intensity of water quality impairments such as harmful algal blooms (HABs)[2,3] and hypoxia[4,5], which impact human and ecosystem health[6,7] and are also linked to increased ocean acidification[8-10].

Anthropogenic activities that directly impact nitrogen export to water bodies include nitrogen inputs to watersheds[11-19] and conversion of land to cropland and developed land[20,21]. These activities are themselves linked to societal trends in population, wastewater management, fossil fuel use, agricultural management and productivity, dietary choices, and import and export of food products. As societies change, their choices will therefore also impact nitrogen runoff[22-30], but the possible impact of these changes have only been examined on limited scales[24,25,28,29] and out to at most 2030[22,27] or 2050[23,26].

Anthropogenic activity also indirectly impacts nitrogen export through climate change and associated changes in precipitation patterns, because precipitation amount, intensity, and frequency affect the amount of nitrogen transported to downstream water bodies[14,16,19,31]. Future changes in precipitation patterns will depend on broader climate outcomes[32,33], thus impacting future nitrogen loads, but existing studies have primarily been conducted at watershed-scale, and have only considered precipitation projections averaged across an ensemble of global climate models (GCMs)[34-37] or projections from one to three individual GCMs[16,38-40]. One recent study, however, used precipitation projections from 21 GCMs to examine changes in total nitrogen (TN) export for the entire continental United States, finding that changes in precipitation patterns alone could increase riverine TN export for the continental United States by 19% by the end-of-the-century for a business-as-usual scenario[41].

Only a handful of studies has examined the combined effects of land management and climate change on nitrogen loading, and these have focused on individual watersheds[42,43].

An assessment of future nitrogen loading under various land use and land management scenarios, on their own and in combination with concomitant changes in climate, is necessary to understand the impacts of long-term societal changes on nitrogen loading. Here, we assess these impacts for the continental United States and through to the end of the 21st century across a set of alternative socioeconomic development pathways.

Projected future changes due to land conversion to cropland or developed land and changes in fertilizer application rates as considered here are based on integrated land use and emission scenarios obtained by combining shared socioeconomic pathways (SSPs, a.k.a. socioeconomic development pathways) and representative concentration pathways (RCPs, a.k.a. climate outcomes)[44]. Five socioeconomic development pathways were developed in preparation for the Coupled Model Intercomparison Project Phase 6 (CMIP6) set of climate simulations[45], and include SSP1 Sustainability, SSP2 Middle of the road, SSP3 Regional rivalry, SSP4 Inequality, and SSP5 Fossil-fueled development[46]. These five SSPs represent state-of-the-art prototypical pathways for global environmental change during the 21st century, including projections under assumptions ranging from sustainable growth to increased consumption over the period 2015–2100. These SSPs, however, are not intended to represent a comprehensive set of possible futures, and other pathways have also been proposed[47-49]. Integrated assessment models (IAMs) were used to produce scenarios by combining these pathways with a range of climate outcomes[44]. The resulting land use projections were harmonized so as to provide a common starting point in 2015, and are available through the land use harmonization 2 (LUH2) dataset[50].

A total of six combinations of socioeconomic development pathways and climate outcomes are considered here. These include the four Tier 1 scenarios[44], namely SSP1-2.6, SSP2-4.5, SSP3-7.0, and SSP5-8.5, each of which was contributed by a different IAM (Supplementary Table 1), as well as two of the Tier 2 scenarios, namely SSP4-3.4 and SSP4-6.0, both of which were contributed by the GCAM IAM[51] (see Methods for additional details). Following earlier work[44], the second number in each scenario represents the corresponding climate outcome; for example, SSP1-2.6 is the SSP1 Sustainability pathway combined with the RCP2.6 climate outcome (i.e., achieving a radiative forcing of 2.6 W/m² in 2100). The SSP4 pathway was combined with two climate outcomes, yielding the SSP4-3.4 and SSP4-6.0 scenarios, achieving a radiative forcing of 3.4 and 6.0 W/m² in 2100, respectively.

In addition, examining the combined impacts of socioeconomic development pathways and changes in precipitation patterns associated with a particular climate outcome can provide insight into whether direct and indirect impacts of anthropogenic activity will compound or offset one another in terms of their effects on nitrogen loading. We therefore also examine the joint influence of direct (via land use and land management) and indirect (via climate change-induced changes in precipitation patterns) societal impacts on future TN loading. Future changes in precipitation patterns are based on projections from CMIP5 models[52] for four climate outcomes (RCP2.6, RCP4.5, RCP6.0, and RCP8.5) (see Methods for additional details). Based on previous work[19,41], future changes in precipitation that are considered include changes in total annual precipitation and changes to extreme springtime precipitation.

The impact of changes in land use, land management, and precipitation patterns on TN flux is examined at eight-digit hydrologic unit (HUC8) subbasin scale within the continental United States using an updated version of an empirical model[19] that estimates TN fluxes based on annual precipitation, extreme springtime precipitation, net anthropogenic nitrogen inputs, and land use (see Methods for further details).

We find that societal choices will substantially impact riverine total nitrogen loading for the continental United States by the end of the century, and that regional impacts will be even larger. Future changes in precipitation will exacerbate loading increases. Although increased loading is possible for both high- and low-emission pathways, we find that limiting climate change and eutrophication can also be achieved concurrently. Increases in cropland area and agricultural intensification will likely impact other parts of the world as well, including vast portions of Asia.

## Results and Discussion

**Impact of changes in land use and land management**. We find that future societal choices about land use and land management will have dramatic impacts on riverine TN loading; across the six prototypical scenarios examined here, impacts range from an increase of 54% (SSP4-3.4 Inequality (3.4)) to a decrease of 7% (SSP5-8.5 Fossil-fueled development) by the end-of-the-century for the continental United States (Figs. 1 and 2a) in the absence of changes in precipitation patterns. Differences in riverine TN loading depend on the scales of agricultural production and anthropogenic nitrogen input, which in turn depend on global demand for food, demand for bioenergy, assumptions about trade, and assumptions about agricultural management practices. Demand for food depends on global population, income, and diet, while bioenergy depends on the degree of climate mitigation and the cost and availability of mitigation options. For the continental United States, the SSP4-3.4 scenario, which assumes low emissions, high climate mitigation, and moderately high demand

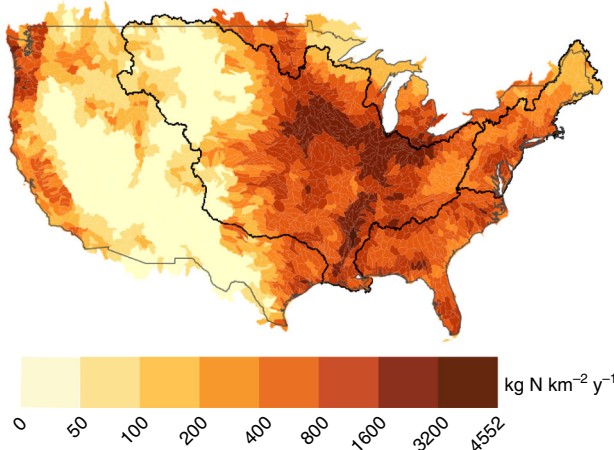

**Fig. 1** Mean total nitrogen flux for watersheds in the continental United States for the historical period (1976–2005), averaged across 30 years and 21 CMIP5 models. The black outlines highlight the Mississippi Atchafalaya River Basin (MARB) and the Northeast region. Net anthropogenic nitrogen input used in these estimates was estimated based on the observational record

for domestic food production, would increase TN loading by 54% (Fig. 2, Supplementary Figure 1b), due primarily to increased production of bioenergy, and associated fertilizer application[51]. Similarly, the projected increases in TN loading for the SSP3-7.0 Regional rivalry scenario is also due to an expansion of agricultural land in the United States, but this time resulting from rising demand for domestic food production[53]. For the SSP4-6.0 Inequality (6.0) scenario, the projected increase in TN loading arises from a high demand for both food and bioenergy production, with the latter being lower than for the SSP4-3.4 scenario[51]. In contrast, the SSP5-8.5 scenario associated with very high greenhouse gas emissions would lead to a modest decrease in future TN loading of 7%, attributable to a small growth in population in the United States and no climate mitigation efforts, resulting in lower agricultural demand and fertilizer use. Any increase in food demand within the United States would be met by livestock intensification, improvements in fertilizer use efficiency, and increased import of food[54]. Overall, the majority of examined scenarios are associated with increased TN loading, and the magnitudes of these increases outweigh the decreases for the remaining scenarios. The most dramatic projected changes would occur for the end-of-the-century period (2071–2100), although changes ranging from an increase of 16% to a decrease of 13% are also projected by the middle-of-the-century (2031–2060; Supplementary Figures 1a and 2).

Overall, we find that some scenarios that are beneficial in terms of reducing greenhouse gas emissions (e.g., SSP1-2.6 and SSP4-3.4) may result in massive increases in nitrogen loading, pointing to exacerbated eutrophication as an unintended consequence of climate mitigation. In contrast, high-emission scenarios (e.g., SSP5-8.5) could lead to modest amelioration of eutrophication within the continental United States. Other scenarios outside of the range considered here are of course also possible[45,55]. This finding points to a potential trade-off between goals of mitigating climate and eutrophication; however, climate change mitigation can also be achieved in more sustainable ways that reduce the unintended consequences of environmental degradation[56]. To some extent this is achieved in the SSP1-2.6 scenario as implemented by the IMAGE IAM[57] (Fig. 2, Supplementary Figure 1a, b), which demonstrates that aggressive climate mitigation can in principle be achieved with minimal increases

in nutrient loading. The remaining scenarios, which are associated with a large intensification of domestic food production (SSP4-6.0 and SSP3-7.0) achieve neither goal, with little climate mitigation (i.e., high-fossil fuel consumption) but also substantial potential increases in eutrophication. Overall, these scenarios demonstrate that the benefits of climate change mitigation strategies should be assessed in a manner that also accounts for their impacts on eutrophication and the sustainability of water resources more broadly, such that climate mitigation strategies can be developed that limit unintended environmental impacts.

The possible magnitude of changes to TN loading resulting from the land use and land management scenarios considered here (up to 54% by the end-of-the-century) is even greater than the magnitude of possible TN loading increases due to future changes in precipitation (up to 19% by the end-of-the-century) reported in an earlier study[41]. In addition, the range of possible changes across the six scenarios considered here (+54 to −7%), which can be thought of as an approximate measure of uncertainty in future TN loading changes resulting from anthropogenic activity within the continental United States, is also substantially larger than the range across future climate outcomes (+11% to +19%[41]). This comparison indicates that changes to the physical climate system must be considered when evaluating risks associated with future eutrophication, but that anthropogenic activity within the affected regions (via land use and land management) has an even stronger potential influence on loading.

How society will evolve in the future is uncertain, even under a common set of high-level assumptions about economic futures and population dynamics. As a result, the assumptions prescribed for each societal pathway manifest themselves differently as a function of the IAM used to represent the pathway (Fig. 3). In the LUH2 dataset, the scenarios for the SSP4-3.4 and SSP4-6.0 scenarios are based on simulations from the GCAM model[51], while the four other scenarios are based on four other IAMs (Supplementary Table 1). Examining differences in scenarios generated by different IAMs for a given combination of socioeconomic development pathway and climate outcome gives a qualitative sense of both the uncertainty associated with societal response under given assumptions, and the model uncertainty associated with the use of any particular IAM. With this goal in mind, here we compare total fertilizer use projections from the LUH2 database to projections for all six scenarios based on the GCAM model. We do not compare impacts on TN loading directly, because spatially explicit and harmonized GCAM outputs are not available for all six scenarios; however, the differences in fertilizer use (which stem from changes in land use and changes in fertilizer application rate) would result in corresponding differences in loading if the requisite data were available. The GCAM model implementation of the SSP1-2.6 and SSP2-4.5 scenarios result in much larger cropland area and fertilizer usage relative to the corresponding marker scenarios in the LUH2 database, because the GCAM scenario assumes a much stronger reliance on biofuels. More broadly, for the four SSPs for which GCAM was not used as the LUH2 marker scenario, the average magnitude of the difference in fertilizer usage in 2100 between the GCAM implementation of a given scenario and the corresponding LUH2 marker scenario is 8.1 Tg N, which is greater than the full spread across the LUH2 database for these four scenarios (7.0 Tg N, Supplementary Table 1). Recently released global fertilizer usage estimates for the five SSPs used here based on the IMAGE IAM also illustrate the differences implied by the choice of IAM[30]. These large differences across IAMs highlight the uncertainty in societal outcomes even under a common set of driving assumptions.

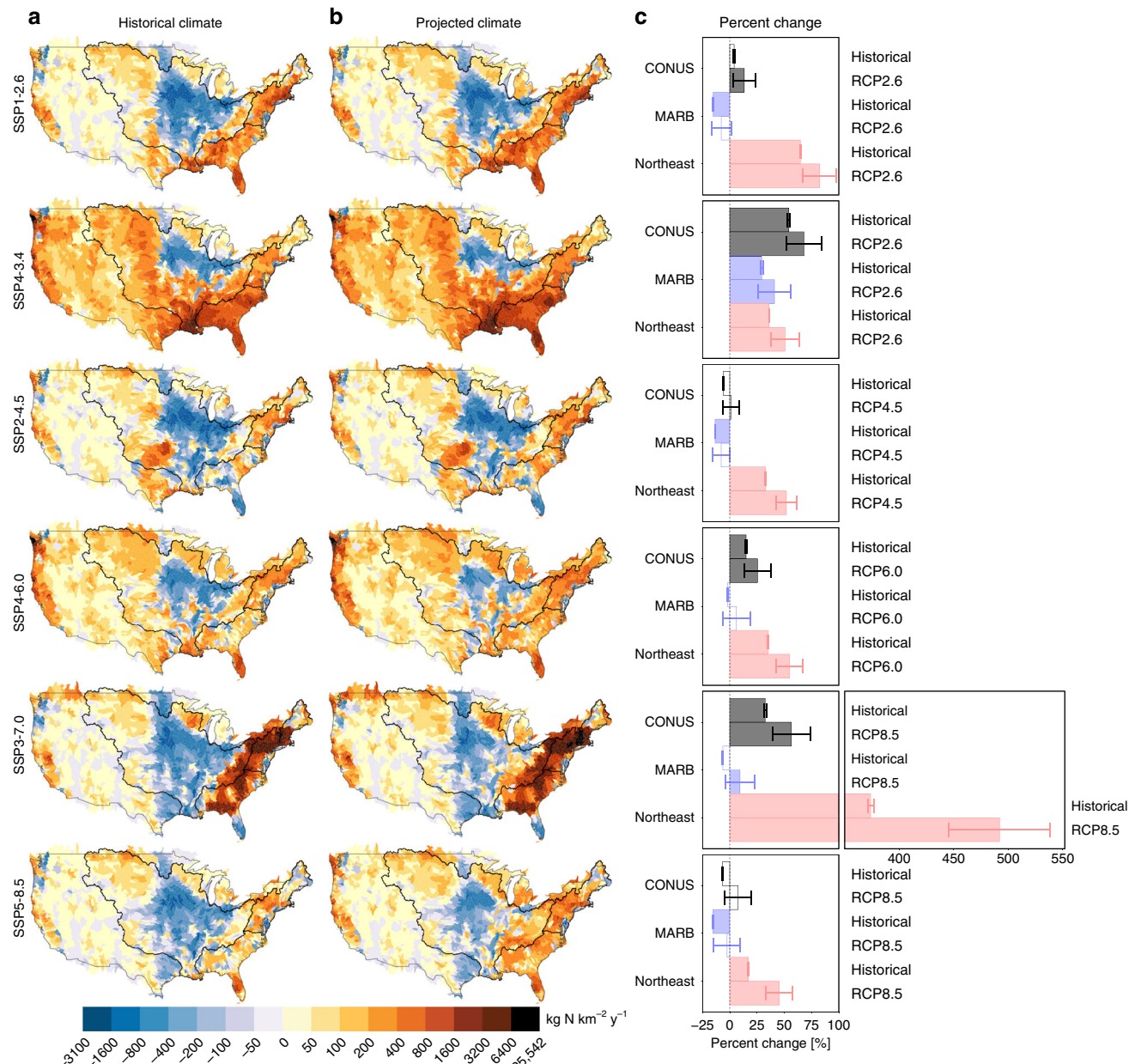

**Fig. 2** The range of socioeconomic pathways considered here result in large differences in future nitrogen loading, both with and without concomitant changes to climate. Change in mean total nitrogen flux by the end-of-the-century (2071–2100) relative to the historical period (1976–2005) for the SSP1-2.6, SSP4-3.4, SSP2-4.5, SSP4-6.0, SSP3-7.0, and SSP5-8.5 scenarios based on projected land use and fertilizer application rates, both without (**a**) and with (**b**) impact of concomitant changes in total annual and springtime extreme precipitation. Precipitation changes are based on RCP2.6 for SSP1-2.6 and SSP4-3.4, RCP4.5 for SSP2-4.5, RCP6.0 for SSP4-6.0, and RCP8.5 for SSP3-7.0 and SSP5-8.5. **a**, **b** show projected change at HUC8 watershed scale, while **c** shows percentage change, both without (top bar) and with (bottom bar) concomitant changes to precipitation, for the continental United States (CONUS), the Mississippi Atchafalaya River Basin (MARB), and the Northeast, with regions as outlined in **a** and **b**. Filled bars represent a robust change and error bars represent one standard deviation. Historical net anthropogenic nitrogen input was estimated based on the observational record, while future input was estimated using data from the LUH2 dataset

The impacts of societal pathways on nitrogen loading are highly variable across regions, with the relative magnitude of some of the regional impacts dwarfing those observed at the continental scale (Fig. 2). For instance, all six scenarios would result in a robust increase in loading for the Northeast region by the end-of-the-century relative to the historical period, with increases ranging from 17% (SSP5-8.5) to 374% (SSP3-7.0). Conversely, three of the scenarios that would result in increased loading for the continental United States by the end-of-the-century would actually result in a decreased loading within the Mississippi-Atchafalaya River Basin (MARB), namely the SSP1-2.6, SSP4-6.0, and SSP3-7.0 scenarios (Fig. 2). The projected decrease in future MARB loading relative to the historical period (1976–2005) is due to a decrease in cropland area between 1997 and 2015 that is not fully offset by land use changes through to the middle-of-the-century and end-of-the-century periods (Supplementary Figure 3). The scale and heterogeneity in regional impacts indicates that impacts of broader societal pathways may

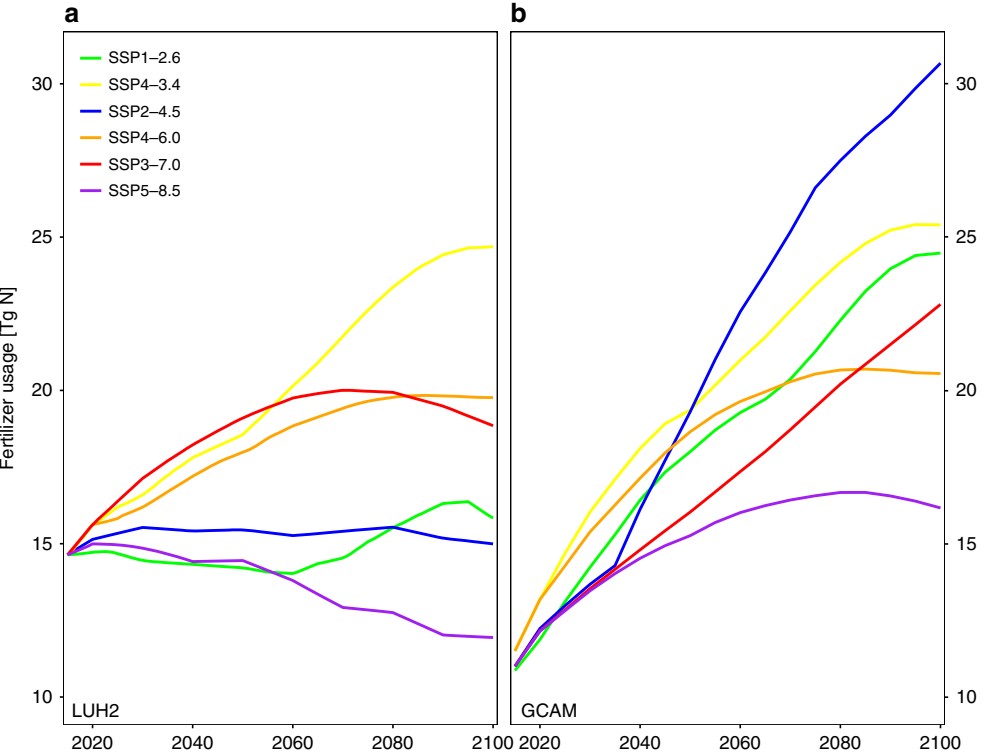

**Fig. 3** Different integrated assessment models lead to dramatically different interpretations of the socioeconomic scenarios considered here. Time series of projected fertilizer usage for the continental United States for the six scenarios based on **a** the LUH2 dataset and **b** the GCAM integrated assessment model. In the LUH2 dataset, the SSP4-3.4 and SSP4-6.0 marker scenarios were contributed by the GCAM model, while the SSP1-2.6 scenario was contributed by IMAGE model, the SSP2-4.5 scenario by the MESSAGE-GLOBIOM model, the SSP3-7.0 scenario by the AIM model, and the SSP5-8.5 scenario by the REMIND-MAGPIE model

be highly geographically uneven, with clear winners and losers in terms of impacts on eutrophication. In the case of the Northeast and the MARB, both of these regions already have high historical loading (Fig. 1), such that any long-term trends are likely to have important consequences on regional water quality. The risks and uncertainty associated with regional differences in impacts are further compounded by differences across IAMs in their representations of specific scenarios (Fig. 3).

We further find that change in cropland area, developed area, and fertilizer application rate are all important contributors to future changes in TN flux (Fig. 4, Supplementary Figure 4). Figure 4 reproduces the information from Fig. 2c with historical climate, but also illustrates the changes in loading resulting only from changes to cropland area (and associated fertilizer application), only from changes to developed land area, and only from changes in fertilizer application rates. For most scenarios and regions, more than one of these factors contributes substantially to the overall impact. As described earlier, the primary driver of the projected changes to cropland area is increased biofuel production for the SSP1-2.6 and SSP4-3.4 scenarios[51,57] (Supplementary Figure 5), and increased food demand for the SSP3-7.0 and SSP4-6.0 scenarios[51,53]. The primary drivers of projected changes to developed land area are population growth and rates of urbanization. For fertilizer application rates, the primary drivers mirror those of changes to cropland area.

**Impact of concomitant changes in precipitation.** The discussion up to this point has focused on the impacts of changes in land use and land management in response to broader societal shifts, as represented by the LUH2 scenarios and additional scenarios from the GCAM IAM. We now shift to accounting for the compound impacts of changes in land use and land management as well as changes in precipitation patterns resulting from climate change.

Concomitant climate-induced changes in precipitation can either amplify TN flux increases or negate TN flux reductions due to land management, and the impacts of land use and land management and those of climate are not necessarily simply additive (Fig. 2). For example, the reductions in TN fluxes for the continental United States resulting from anticipated changes in land use and land management for the SSP2-4.5 and the SSP5-8.5 (Fig. 2) scenarios would be fully offset by the impacts of projected changes in precipitation patterns. For the four other scenarios, the increases in TN fluxes would be further amplified by projected changes in precipitation patterns. As expected, the additional impact of climate would be greatest for scenarios associated with more extensive changes to climate (SSP3-7.0 and SSP5-8.5). Taken together, whereas the range of anticipated changes in the absence of concomitant changes to climate is −7 to 54% for the continental US, this range becomes 1–68% when both sets of drivers are taken into account. In addition, as previously noted[41], the impacts of climate-change-induced precipitation changes are themselves regionally heterogeneous. The Northeast exhibits the strongest impacts both when considering land management alone (with loading increases ranging from 17 to 374%) and when considering the additional impacts of climate change (with total loading increases from both drivers ranging from 45 to 492%). In addition, in some cases the impacts of land management and climate changes are not additive (e.g., SSP3-7.0 scenario in the Northeast; Supplementary Figure 1f), illustrating the importance of considering these two sets of drivers concurrently. It is

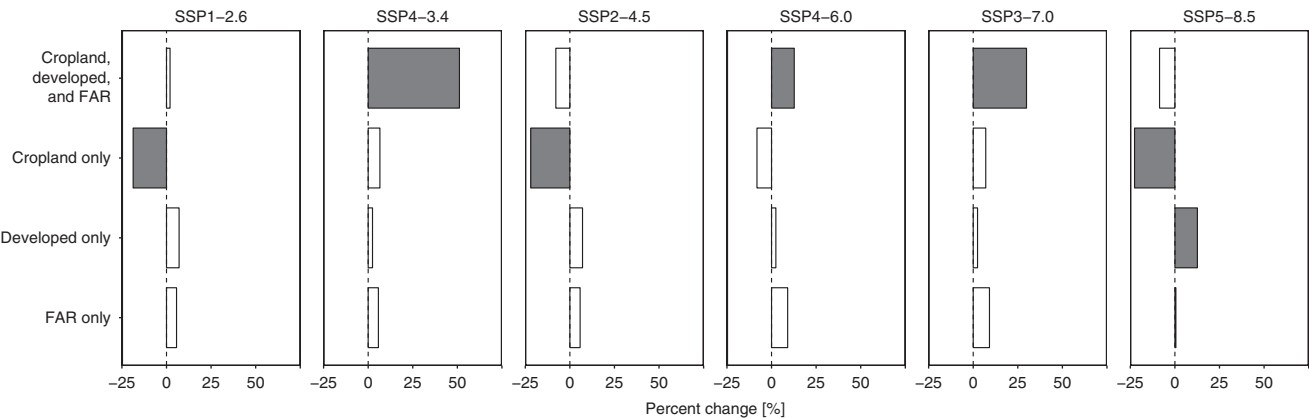

**Fig. 4** Changes in cropland area, developed area, and fertilizer application rate all contribute to changes in future nitrogen loading. Percentage change in mean total nitrogen flux by the end-of-the-century (2071–2100) relative to the historical period (1976–2005) for the six examined scenarios based on projected cropland, developed land, and fertilizer application rates, projected cropland but historical developed land and fertilizer application rates, projected developed land but historical cropland and fertilizer application rate, and projected fertilizer application rates (FAR) but historical land use. Precipitation is kept at historical levels. The projected changes are shown for the continental US (CONUS). Filled bars represent a robust change. Due to the nonlinear form of the model and the impact of interactions among changes in land use and net anthropogenic nitrogen input, the impacts of the three individual components are not simply additive. Net anthropogenic nitrogen input was estimated based on the LUH2 database for both the historical period and future period

noteworthy that for some scenarios such as SSP2-4.5, decreases in nutrient loading due to changes in land management can help to offset increases in loading that would result from changes in precipitation patterns associated with the corresponding climate outcome. Such examples point to the possibility of not only limiting impacts of land management change on nutrient loading (as discussed earlier), but indeed of using land management to address eutrophication resulting from changes to the physical climate system.

Beyond the United States, Asian regions that have previously been identified as being particularly susceptible to increases in eutrophication as a result of future changes in precipitation[41] also exhibit large projected future increases in cropland area and total fertilizer usage across the majority of prototypical pathways examined here (Fig. 5, Supplementary Figure 6), suggesting that the overall scale of direct and indirect anthropogenic impacts is likely to be substantially larger than when considering climate alone. Globally, among regions with nonzero fertilizer use during the historical period (colored land regions in Fig. 5a), the direction of change in fertilizer usage is consistent for at least five of the six examined scenarios for 67% of land area (colored regions in Fig. 5b). Regions in South, East, and Southeast Asia stand out relative to other regions in having both high-historical fertilizer usage and a robust and large projected increase in fertilizer use, placing them at a high risk for a substantial future increase in eutrophication. These regions also coincide with those identified in an earlier study[41] as being at high risk for future increases in eutrophication based on their high-historical precipitation and robust projected future increase in precipitation. When taken together, these two sets of analyses indicate that climate and societal changes are likely to conspire in this region to substantially impact inland and coastal water quality. Such intensified eutrophication could substantially impact water supplies, fisheries, and ecosystems. It is important to note, however, that the prototypical pathways examined here do not cover all possible outcomes, which will depend in no small part on policy and legislative action by individual countries. For example, future nitrogen use efficiency may improve in China and India as both countries are now increasing efforts to reduce nitrogen pollution by encouraging farmers to adopt best management practices[58,59] and by promoting organic farming

(https://darpg.gov.in/sites/default/files/Paramparagat%20Krishi%20Vikas%20Yojana.pdf).

Overall, we find that societal choices about land use and land management will have dramatic impacts on future nitrogen loading within the continental United States. In some cases, climate mitigation efforts could result in unintended increases in eutrophication, but opportunities to concurrently limit climate change and eutrophication do exist. Climate-induced changes in precipitation will exacerbate any loading increases due intensification of land use, or reduce the benefits of improved land management. Globally, South, East, and Southeast Asia will likely experience increased eutrophication by the end-of-the-century given projected trends in land use, land management, and precipitation across the majority of scenarios examined here. These results imply that strategies aimed at reducing eutrophication and associated water quality impacts must be based on an assessment of the combined impacts of changes in socioeconomic development and climate. Whereas some societal pathways can achieve climate mitigation while increasing food production, other mitigation pathways may have significant detrimental impact on environmental sustainability.

## Methods

**Analysis domain and discretization.** The impact of changes in land use, land management, and precipitation patterns were examined for the continental United States at the HUC8 watershed scale. Analogously to an earlier study[41], the analysis included all HUC8 catchments except for ten that are predominately made up of water, yielding a total of 2105 HUC8 watersheds. The HUC8 watershed boundaries were obtained from the U.S. Geological Survey (USGS) Watershed Boundary Dataset (WBD).

**Historical and future precipitation.** Historical and future precipitation was obtained following the methodology from an earlier study[41], and was based on CMIP5 model projections for the RCP2.6 (16 models), RCP4.5 (20 models), RCP6.0 (12 models), and RCP8.5 (21 models) emission scenarios. The precipitation projections were obtained from the Downscaled CMIP3 and CMIP5 Climate and Hydrology Projections archive[60], which are bias-corrected and spatially downscaled to 1/8° resolution. For the model development step (see Section "Empirical model of nitrogen loading"), historical precipitation records from 1987 to 2012 were substituted, as described in an earlier study[41].

**Historical and future land use and fertilizer application.** Projected changes in land use and land management within the continental United States were based on the land use harmonization 2 (LUH2)[50] dataset prepared for the sixth phase of the

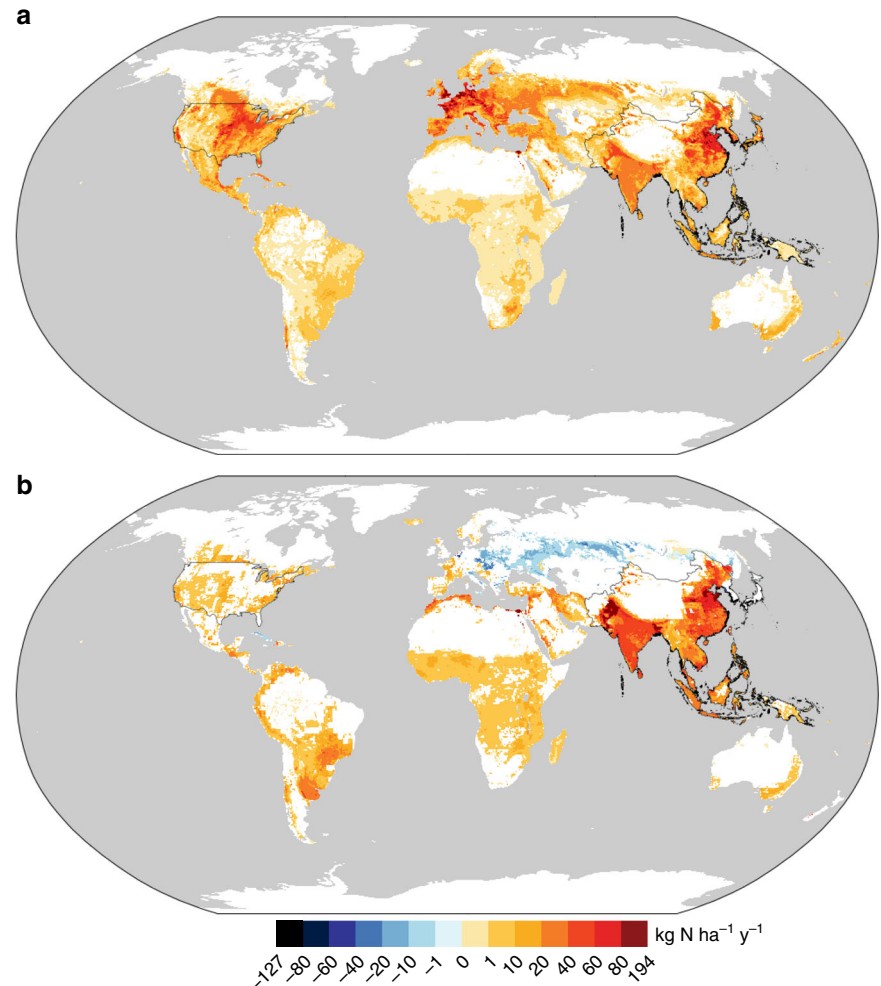

**Fig. 5** Highly populated regions in South, East, and Southeast Asia are at high risk of increased eutrophication due to large and robust projected increase in fertilizer application rates. **a** Fertilizer usage per unit area for the historical period (1976–2005) averaged across the 30 years. **b** Projected change in fertilizer usage per unit area by the end-of-the-century period (2071–2100) relative to the historical period, averaged across the 30 years and six scenarios and shown only for regions where at least five of the six scenarios agree on the direction of change from the historical to the end-of-the-century period. **a** Regions shown in white have no fertilizer usage; **b** regions shown in white either have no fertilizer usage or do not have at least five scenarios with a consistent direction of change

Coupled Model Intercomparison Project (CMIP6). The LUH2 dataset includes land use scenarios for 12 land use categories (Supplementary Table 2) and provides land management scenarios that include fertilizer application rate for five crop types for all global land grid cells at 1/4° spatial resolution and annual temporal resolution. The LUH2 dataset is available for two time periods, with the historical (LUH2 v2h) period covering 850–2015 and the future period (LUH2 v2f) covering 2015–2100. For the future time period, land use and land management scenarios are available for five Shared SSP, each of which is coupled with a corresponding representative concentration pathway (RCP) or climate outcome (SSP1 with RCP2.6, SSP2 with RCP4.5, SSP3 with RCP7.0, SSP4 with RCP3.4, SSP4 with RCP6.0, and SSP5 with RCP8.5), yielding six combined scenarios overall because the SSP4 pathway is coupled to two different RCPs. Although multiple IAMs generated scenarios for each pathway, the scenarios compiled in the LUH2 dataset are based on the marker IAM for each pathway (Supplementary Table 1). Within the LUH2 database, the IAM outputs of various scenarios were harmonized such that the future projections for the six SSPs smoothly connect with the historical projections[50]. This harmonization process results in a shift in the future scenario projections from the original IAM estimates for a portion of the future period.

The SSPs describe prototypical trajectories for societal development that may occur in the future. The range of SSPs is intended to span the uncertainty associated with societal development, although outcomes outside of this range cannot be ruled out. Various assumptions regarding the SSP pathways[46] and the IAM implementation of these pathways are described in earlier manuscripts[51,53–55,57,61] and a brief overview of these assumptions is included here. The SSP1-2.6 Sustainability scenario, contributed by the IMAGE model[62], assumes sustainable growth in the future that focuses on human well-being and respect for environmental boundaries[57]. For the continental United States, cropland area and fertilizer usage are projected to increase slightly by the end-of-the-century, driven by an increase in energy crops. The SSP2-4.5 Middle of the Road scenario, contributed by the MESSAGE-GLOBIOM model[63,64] assumes that future growth patterns will be similar to historical patterns, and is designed to cover the middle ground in terms of mitigation and adaptation challenges between more extreme SSPs[61]. For the continental United States, cropland area is projected to remain unchanged while fertilizer usage is projected to decrease slightly. The SSP3-7.0 Regional rivalry scenario, contributed by the AIM model[65], represents a world with increased nationalism and regional rivalry and decreased globalization[53]. For the continental United States, this scenario leads to increased deforestation associated with an increase in cropland area and fertilizer usage. The SSP4 Inequality pathway represents a world with increasing inequality both within and across countries[51]. This pathway was combined with two climate outcomes, yielding the SSP4-3.4 and SSP4-6.0 scenarios, both based on the GCAM model[66]. These two scenarios are based a common set of high-level assumptions about societal growth, but with different climate outcomes representing different degrees of climate mitigation. For the continental United States, for both the SSP4-3.4 and SSP4-6.0 scenarios, cropland area and fertilizer usage is projected to increase in the future, as a result of an increase in bioenergy crops and an increase in domestic food production. The increase in bioenergy production for the SSP4-3.4, however, is much larger than for the SSP-6.0 (Supplementary Figure 5). The SSP5-8.5 Fossil-fueled development scenario, contributed by the REMIND-MAGPIE model[67], represents a world with increased globalization and high economic growth[54]. For the continental United States, cropland area remains largely unchanged, developed land areas increase slightly, and fertilizer usage decreases due to an increase in productivity.

**Historical and future net anthropogenic nitrogen inputs**. Nitrogen input to the watersheds was captured in the form of net anthropogenic nitrogen input (NANI), which is defined as the sum of five components: fertilizer nitrogen input, agricultural nitrogen fixation, net food and feed import, atmospheric deposition, and nonfood crop export. Net food and feed import accounts for crop and animal nitrogen production (removal of nitrogen from the watershed) and human and animal nitrogen consumption (addition of nitrogen to the watershed) and includes bioenergy production and consumption. The nonfood crop export represents the nitrogen in cotton and tobacco harvested for sale and exported elsewhere for nonfood use. On average during the historical period, fertilizer was +72% of NANI; fixation was +45% of NANI; deposition was+ 14% of NANI; food and feed import was −30% of NANI (negative sign designates net export over the continental United States); and nonfood crop export was+ 0.78% of NANI. The nonfood crop export represents nitrogen leaving the watershed and its value is subtracted from NANI while all other components are added to estimate NANI. NANI was estimated using two different methods. First, for the empirical development and for assessing historical TN loading, available data on the various NANI components was used to estimate historical NANI. Second, for estimating future TN loading, NANI estimates were developed based on output available in the LUH2 dataset. In addition, historical NANI was also estimated directly based on LUH2 data in the analysis focusing on isolating the impact of future changes to cropland area, developed land, or fertilizer application rates, in the absence of concurrent changes to the other factors. The first approach is described here, while the second is presented in Section "Future changes in TN flux due to change in land use and land management".

Observation-based NANI estimates were obtained as described in an earlier study[19] for 1987–2005. NANI for 1976–1986 was estimated by modifying this methodology to accommodate a more limited data record during this decade. Briefly, fertilizer rates for 1987–2005, 1985–1986, and 1976–1984 were obtained from three earlier studies[68–70], respectively. Atmospheric deposition rates for 1979–2005 were based on data from the National Atmospheric Deposition Program[71]. The NADP monitoring network was established in 1978, but had very few monitoring stations in that initial year. Atmospheric deposition rates for 1976, 1977, and 1978 were therefore assumed remain constant at 1979 levels. Agricultural nitrogen fixation, net food and feed import, and nonfood crop export were estimated for the agricultural census years of 1987, 1992, 1997, 2002, and 2007 using the NANI toolbox[72,73] and for the intervening years (1988–1991; 1993–1996; 1998–2001; 2003–2006) using linear interpolation. For the years preceding 1987, agricultural nitrogen fixation and net food and feed import were estimated for each HUC8 watershed by extrapolating a linear trend estimated based on the values for 1987, 1992, 2997, 2002, 2007, and 2012. Nonfood crop export, which are an extremely small component of NANI (on average 0.78% of NANI), were assumed to remain at 1987 levels for 1976–1986.

**Empirical model of nitrogen loading**. The empirical model reported in an earlier study[19] that relates precipitation, land use, and NANI to TN loading at the HUC8 scale was developed based on land use categories included in National Land Cover Database 2006 (NLCD 2006). There is, however, no simple correspondence between the NLCD 2006 land use categories and those reported in the LUH2 dataset. Rather than create an approximate mapping, we instead developed a modified model, with model selection and model calibration based directly on the land use variables available in the LUH2 dataset. The approach was analogous to the earlier study[19], with the following differences. The 12 land use categories of LUH2 dataset were aggregated into six categories as illustrated in Supplementary Table 2. Following the methodology of the earlier study[19], all six land use categories and all their combinations were considered in the model selection. In addition, because the observation-based NANI data are now available through 2012[72,73], the calibration dataset included the same catchments as the earlier study[19], but observed TN loading for 2012 was added where available. The total number of catchment-years for model build was thus increased from 242 to 280. Also, whereas the NLCD database only provides land use for a single year, the LUH2 dataset provides annual values for all land use categories, and land use was therefore no longer held constant over time.

The final model based on LUH2 land use categories represents the natural log of TN loading ($\ln(Q_{TN})$) as:

$$\ln(Q_{TN}) = 0.538 + 0.438 \cdot f_{NANI} + 0.0012 \cdot P_{Annual} + 0.0033 \cdot P_{MAM,p>0.95} + 0.0213 \cdot LU_{D,C},$$
(1)

$$f_{NANI} = a\sinh(NANI/2) = \ln\left(NANI/2 + \sqrt{(NANI/2)^2 + 1}\right),$$
(2)

where $f_{NANI}$ [ln(kg N km$^{-2}$ y$^{-1}$)] is the transformed annual NANI for the target year; $P_{Annual}$ [mm] is annual precipitation in the catchment; $P_{MAM,p>0.95}$ [mm] is extreme precipitation in March–May expressed as the amount of precipitation that fell on days with precipitation greater than the 95th percentile (where the percentiles are calculated based on daily precipitation for 1981–2010); $LU_{D,C}$ [%] is the percentage of the catchment area classified as developed or cropland. While the original model[19] explained 76% of the observed variability in $\ln(Q_{TN})$, the updated model explained 67%, suggesting that the land use categories in the LUH2 dataset

are less representative relative to those in the NLCD data. Beyond the land use description, the updated model is very similar to the original, with the selected nonland-use predictor variables being identical to the original ($f_{NANI}$, $P_{Annual}$, and $P_{MAM,p>0.95}$). In terms of land use, the two variables based on NLCD land use classification, namely percent land cover by wetlands and percent land cover by forests or shrublands are replaced by a single variable based on the LUH2 land use classification, namely percent agricultural or developed land use. The two land use variables included in the original model were associated with negative drift coefficients, meaning that, all else being equal, an increase in wetlands, forests, or shrublands was associated with a decrease in TN loading. In the updated model, croplands and developed land is associated with a positive drift coefficient, meaning that, all else being equal, an increase in croplands and developed lands leads to an increase in TN loading. Therefore, although the two models use different land use variables, they tell an entirely consistent story about the relative impacts of various land use categories on TN loading. Mechanistically, increased TN loading from agricultural and developed land (beyond the impact accounted for by NANI) is attributable to drainage that directly transports TN from land to streams and the lack of nitrogen retention that occurs in more natural landscapes.

**Historical TN flux estimation**. TN fluxes were estimated for the historical period (1976–2005) by applying the empirical model in Eq. (1) to all HUC8 watersheds within the continental United States. For the historical period we utilized observation-based NANI estimates (see Section "Historical and future NANI datasets"), land use based on historical LUH2 dataset (see Section "Historical and future land use and fertilizer application datasets"), and precipitation projections from 21 CMIP5 models (see Section "Historical and future precipitation datasets"). This approach is analogous to that used in an earlier study[41], except with the updated datasets and empirical model as described in Section "Datasets used".

**Changes due to change in land use and land management**. The impact of future changes in land use and land management on TN flux was estimated using land use (Supplementary Figure 3) and NANI (Supplementary Figure 7) based on the six scenarios from the LUH2 dataset (as described below), and historical precipitation based on 21 CMIP5 models (see Section "Historical and future precipitation datasets"). The robustness of projected change in TN flux was assessed in a manner analogous to an earlier study[41], whereby a change is considered robust if more than 50% of the models show a significant change, as assessed by a two-sided $t$ test at the 95% confidence level, and more than 80% of the models agree on the sign of change[74]. Because we are holding precipitation at historical levels, the only variability across the CMIP5 models for a given future land use and land management scenario therefore comes from how the historical precipitation patterns for a given CMIP5 model manifest themselves as TN loading in the framework of a given land use and land management scenario. In other words, a change in land use in a particular location of the country may have a different impact within the context of historical precipitation patterns as represented by one CMIP5 model vs. another.

For the future period, three of the NANI components (fertilizer nitrogen input, agricultural nitrogen fixation, and net food and feed import) were estimated based on the future LUH2 dataset for the six scenarios (Supplementary Figure 8). The LUH2 dataset provides fertilizer application rates for five different crop types (C3 annual crops, C4 annual crops, C3 perennial crops, C4 perennial crops, and C3 nitrogen-fixing crops). Fertilizer usage was obtained by summing the area-weighted fertilizer application rate for all crop types. The fertilizer usage was scaled such that estimates from 1987 to 2012 period (obtained from LUH2 v2h) matched existing estimates of fertilizer usage[68] for each HUC2 (first level of hydrologic unit) region (see Supplementary Figure 1 in ref. [41]). Nitrogen fixation was obtained by multiplying the C3 nitrogen-fixing crop area with a constant nitrogen fixation rate of 14,300 kg N km$^{-2}$ y$^{-1}$, which itself was estimated by multiplying the fraction of the C3 nitrogen-fixing crop area for 2007 from the LUH2 database by a fixed nitrogen fixation rate such that the TN fixation at the continental United States scale matched the 2007 nitrogen fixation estimates of 6.9 Tg N based on the NANI toolbox[72,73]. Future food and feed import, which includes bioenergy production and consumption, was estimated using future fertilizer nitrogen input and agricultural nitrogen fixation for each HUC2 region under the assumption that the relationship between food and feed import and fertilizer input and agricultural nitrogen fixation will remain constant over time. The estimate was based on a multiple linear regression with historical food and feed import as the dependent variable, and historical fertilizer nitrogen input and historical agricultural nitrogen fixation as the independent variables (Supplementary Figure 9). Data on historical values for all three variables for 1987, 1992, 1997, 2002, 2007, and 2012 were obtained from the NANI toolbox[72,73]. Future food and feed import is estimated as a function of fertilizer usage and agricultural nitrogen fixation; therefore, reductions in fertilizer usage resulting from either improvements in nitrogen use efficiency or reductions in cropland area will also impact the nitrogen embedded in net food and feed import. The fourth NANI component, atmospheric deposition was estimated based on existing future NOx deposition estimates[75] for the RCP2.6, RCP4.5, RCP6.0, and RCP8.5 emission scenarios, which are based on the community multiscale air quality (CMAQ) model, while the TN flux empirical model was developed by utilizing the National Atmospheric Deposition Program (NLCD) NOx deposition estimates. The future NOx deposition estimates were

therefore adjusted based on scaling factor between CMAQ model based NOx deposition (obtained from https://archive.epa.gov/amad/archive-amad/web/html/wdtdata.html) and NADP-based NOx deposition for 2002, 2003, 2004, 2005, and 2006. The fifth NANI component, nonfood crop export, is an extremely small component of NANI and was held constant at the 2007 value (i.e., the approximate value at the end of the historical period) for all years after 2007.

**Role of changes in cropland, developed land, and fertilizer use**. Future land conversion to (or from) cropland or developed land impacts TN flux via the land use term in Eq. (1) ($LU_{D,C}$), but land conversion to (or from) cropland also indirectly impacts TN flux via the NANI term in Eq. (1) ($f_{NANI}$) because it affect three of the NANI components, namely fertilizer usage, agricultural nitrogen fixation, and net food and feed import. Future changes to fertilizer application rate for a given crop type, on the other hand, impact TN flux only via their impact on the NANI term in Eq. (1) and this only via two of the NANI components, namely fertilizer usage and net food and feed import. To assess the contribution of changes in future cropland (together with its embedded change in anthropogenic nitrogen inputs), developed land, or future fertilizer application rate individually, we developed three additional sets of future TN flux estimates based on: first, future cropland change (and concomitant changes to NANI) but historical developed land and fertilizer application rates; second, future changes to developed land area, maintaining cropland and fertilizer application rates at historical levels; and third, historical cropland and developed land but with future fertilizer application rates. Because the observation-based NANI estimates (see Section "Historical and future NANI datasets") do not make it possible to assess fertilizer application rates for the five crop types in the LUH2 database, for these additional runs the historical fertilizer application rate was based on LUH2 data, and for consistency all other components of NANI were based on LUH2 data (as described in Section "Future changes in TN flux due to change in land use and land management") rather than observation-based NANI. Also, due to the nonlinear form of the model and the impact of interactions among changes in land use and NANI, the impacts of the three individual components are not simply additive.

**Changes due to concomitant changes in precipitation patterns**. The combined impact of changes in land use and land management and changes in precipitation patterns on future TN flux was examined by estimating TN fluxes for future periods based on LUH2-dataset-derived NANI and land use (six scenarios), and precipitation projections based on CMIP5 models (four climate outcomes). Bias-corrected and downscaled (1/8°) precipitation projections were obtained from the CMIP5 archive[60], which includes 21, 16, 20, 12, and 21 models for the historical, RCP2.6, RCP4.5, RCP6.0, and RCP8.5 climate outcomes, respectively. We assessed the joint influence of direct and indirect impacts by combining the six scenarios with the most similar climate outcome available from CMIP5 simulations. The SSP scenarios for which the equivalent concentration pathway exists in CMIP5 were paired. As such, SSP1-2.6 was combined with precipitation based on RCP2.6, SSP2-4.5 was combined with RCP4.5, SSP4-6.0 was combined with RCP6.0, and SSP5-8.5 was combined with the high-emission RCP8.5. The SSP scenarios for which a corresponding concentration pathway is not available within the CMIP5 simulations were paired with the most similar available pathway. Thus, the SSP4-3.4 scenario was paired with RCP2.6. The SSP3-7.0 scenario is described as combining "high societal vulnerability with high forcing"[44], and was therefore paired with the high emission RCP8.5 rather than with the medium stabilization RCP6.0[76].

**TN flux estimation at aggregated scales**. In order to understand TN fluxes for large regions, TN flux estimates for HUC8 watersheds within those regions were combined in a manner analogous to earlier studies[19,41]. Briefly, TN fluxes for large aggregated regions were estimated by aggregating the nitrogen loads [kg N y⁻¹] for all HUC8 watersheds falling within the region. The TN load from a watershed was obtained by multiplying the loading per unit area ($Q_{TN}$ [kg N km⁻² y⁻¹]) by the watershed area ([km²]).

**Differences across scenarios vs. across IAMs**. As part of the scenario development process[45,46], socioeconomic development pathways were converted into quantitative projections of energy, land use, and emissions using several IAMs. A single-IAM implementation (a.k.a., marker scenario) was then selected for each scenario for inclusion in the LUH2 database and ultimately for use as part of the CMIP6 effort. The selection was based on the internal consistency of the full set of SSP markers, and the ability of the different models to represent distinct characteristics of the storylines[45]. Examining the implementations of scenarios by other IAMs, however, can provide insights into alternative interpretations of the same storyline. With this goal in mind, we compared the cropland area (results not shown) and fertilizer usage projections for the six scenarios in the LUH2 database with the cropland area projection for those scenarios as implemented by the GCAM model. Comparing the GCAM outputs provide insight into the differences across the scenarios, while the LUH2 outputs represent the differences across both the scenarios and the IAMs, because the scenarios in the LUH2 database are contributed by five different IAMs. By comparing GCAM and LUH2 projections for matching scenarios, one can therefore get insight into the role of intermodel differences.

**Code availability**. Simulation output evaluated in this study are available from the authors upon request.

## Data availability
Data used in this study are freely available online, as listed in Supplementary Table 3.

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

## Acknowledgements

Supported by NSF grant 1313897 (E.S. and A.M.M.) and AGS-1243095 (P.J.L.), and by the U.S. Department of Energy, Office of Science (K.V.C.). We acknowledge the land use harmonization team for their efforts in providing a harmonized set of land use scenarios.

We acknowledge the World Climate Research Programme's Working Group on Coupled Modelling, which is responsible for CMIP, and we thank the climate modeling groups (listed in Supplementary Table 4 of this paper) for producing and making available their model output. For CMIP the U.S. Department of Energy's Program for Climate Model Diagnosis and Intercomparison provides coordinating support and led development of software infrastructure in partnership with the Global Organization for Earth System Science Portals.

## Author contributions

E.S. and A.M.M. designed the study, performed the research, and analyzed the data with input from K.V.C. and P.J.L. All authors contributed to the interpretation of results and writing of the manuscript.

## Additional information

**Competing interests:** The authors declare no competing interests.

