## [Peer Review File · Nature Communications]

Reviewers' comments:

Reviewer #1 (Remarks to the Author):

The paper provided a global scale assessment of the combined effects of projected climate change and land use on eutrophication. As far as I am aware, this is the first study to do so, and as such this makes a novel contribution to the literature which will be of interest to others in the community and the wider field of environmental sciences. The methodology used to analyse each scenario builds on earlier published work and is extremely well documented and detailed, is in my opinion valid and appropriate, and is rigorous in its approach. It treats well the uncertainty in climate change projection. The documentation, and the earlier published works, would enable another researcher to repeat the analysis.

The conclusions are original, in that they find that climate mitigation may increase eutrophication, although some pathways allow for limiting both climate change and eutrophication. It is made clear that it is the increased use of biofuels that is the cause of the unintended impact of climate change mitigation. More generally, they also report that future societal choices will have a large impact on eutrophication, which is less surprising.

The specific projections of a range of 1-68% increase in nitrogen loading, which result which climate change impacts on precipitation are also included, are quite startling. This statement should, however, be caveated since it is the range associated with the particular set of scenarios studied, and other scenarios can be envisaged that are not explored in the paper.

The paper explores in detail the effects of different levels of climate change mitigation (low, moderate, high), food production (low, moderate high, and generally associated with population trends), combined with climate change itself. Whilst it makes sense to consider climate change scenarios which match the level of climate change mitigation being modelled, some of the combinations of mitigation and food production are not explored. For example, there is no high mitigation and low food demand scenario; and there is no low mitigation and high food production scenario. The two extremes appear to be high mitigation and moderate food production; and low mitigation with no increase in food production. In particular, with regard to the SSP5-8.5 scenario, it seems that the assumption that is made, explained on p 12, that fertilizer use decreases due to an increase in productivity, seems to be a strong driver of the results - ie. this result is an inevitable consequence of the input assumption which seems optimistic. I would argue that in SSP5, where there is high food demand, it would be more likely that fertiliser use would increase. Indeed, this assumption seems counter-intuitive and it is the output of this scenario that suggests that lack of climate change mitigation would reduce eutrophication. Hence it would be useful to explore the

combination of RCP8.5 with other assumptions about population and food demand. SSP5 is not the only SSP that is generally considered compatible with RCP8.5.

Similarly, the issue of the extent to which climate change mitigation could be achieved without a significant increase in biofuel use, or not, needs to be explored and discussed, and brought out much more. Clearly the authors only had access to SSP1-2.6, but a wide range of other scenarios which achieve stringent mitigation targets have emerged in the past year or so, and whilst I am not suggesting that the authors should simulate these, they should discuss this literature in particular focusing on the amount of biofuel use. A particularly relevant paper here is Grubler A, Wilson C, Bento N, Boza-Kiss B, Krey V, McCollum D, Rao ND, Riahi K, et al (2018). "A Low Energy Demand Scenario for Meeting the 1.5°C Target and Sustainable Development Goals without Negative Emission Technologies." Nature Energy. I think it would be good to bring out more a message that climate change mitigation would need to be done in certain ways in order to avoid unintended side effects. Minimising the use of biofuel would also avoid many of the other undesirable side effects - when used at large scales, competition with food production and biodiversity conservation would occur. Hence there is actually a win win option out there which avoids all of these unintended consequences. At the moment the message of the paper is a little misleading, in that it would be easy to take away a message that policy makers need to choose between climate change and eutrophication, or to balance the two. At the moment the results show that some mitigation scenarios increase eutrophication and that some non-mitigation scenarios decrease it, but it may well be that the reverse is also true.

So, the analysis of the LUH2 marker scenarios needs to be complemented by a discussion of the limitations of the scenario combinations explored, which means that more outcomes are possible than those reported upon, and the authors could speculate on what these might be. The setting of cropland area which are imposed, are strong determinants of the results, for example. How robust are the results of these assumptions and what is the rationale behind them? This needs to be explained in more detail.

In Table S2, a sensitivity analysis is provided which contains fertiliser usage projected by GCAM. These scenarios are, as the authors state, very valuable, but they need to be taken through the full analysis (as done for the LUH2 scenarios) in order to explore the full implications of this uncertainty.

Some aspects of uncertainty (such as uncertainty across CMIP5 models) are well explored, and it would be a pity not to complete the picture in this way.

Finally, the discussion would benefit from considering existing legislation about eutrophication, especially in relation to the Figure 5 for Asia. Are there particular pieces of legislation which would be violated, eg in the US or the EU?

Minor comments:

In the results section, at the beginning it is not immediately clear that the effect of changes in climate are not driving the results at this point. As one reads on, it becomes clear that this part of the discussion relates only to the land use and climate mitigation policy drivers, and that climate change effects are added later. I suggest splitting the results section clearly into two to differentiate between these two sections.

My overall opinion is that with some more calculations and appropriate modification to the discussion and conclusions to incorporate the very important points above, that the paper is suitable for publication in Nature Communications. This is necessary to ensure that the policy relevant messages are truly objective and not skewed by the scenarios that happen to be readily available for analysis. This will be important because this paper will undoubtedly influence thinking in the field.

Reviewer #2 (Remarks to the Author):

Sinha et al., based on an updated empirical model used in previous studies, investigate the effects of future land use and land management on nitrogen loading. They found that, depending on the selected LUH2 land use scenario, end-of-century N loading in the US will increase by up to 54%, including large regional differences. In addition, the authors study the combined effects of changes in land use and precipitation and investigate the isolated effects of changes in cropland area, urban area, and fertilizer rates. Sinha et al. also provide some valuable insights about the substantial differences across integrated assessment models when simulating the same SSP-RCP scenario. The study is well-written and the results highlight the relevance of future land use changes on riverine nitrogen loading, even though at some points I wondered how trustworthy the results are given the limited input data. I find the study worth for publication after some clarifications.

One aspect I find somewhat worrying is that by applying the very wide range of land use projections from IAMs/LUH2, substantial changes and differences across scenarios for all kinds of environmental variables are naturally to be expected, but we actually have no clue which of the projections are

realistic. I understand that the scope of this study is not to investigate the plausibility of the scenarios in detail but could the authors emphasize some of the underlying IAM assumptions (apart from “demand for bioenergy drives cropland and fertilizer increase...”, e.g. the assumed rate of yield improvements, nitrogen use efficiency, food demand in the US, what would be implications of large-scale bioenergy cultivation in SSP4-RCP3.4 on food production?) and why there is such a large discrepancy before and after harmonization (I would not call a difference of 14.6 vs. 11.5 TgN yr⁻¹ in year 2015 “small”). How do the projections fit to what happened in the past few years?

In addition, not being familiar with the model, I find parts of the method section quite difficult to follow. For instance, I do not understand the component “net food and feed import”, can you explain it briefly? Is it the nitrogen taken up by plants and then removed from the system via harvest/grazing (and eventually returned elsewhere –if so where?)? Same for “non food crop export”, why is it assumed to be constant? Shouldn’t this vary (and not be extremely small) with the extent of bioenergy crops? Or is bioenergy part of food and feed? Furthermore, the authors simulate future food and feed import/export as a function of future N fertilization/fixation and the relationship seems to be ok for present-day conditions but will this also hold for the future under changing environmental conditions and crop types? Do the IAMs assume increased nitrogen use efficiency and/or different fertilizer rates for bioenergy crops and food crops and how will this affect simulated food and feed import?

Minor comments:

p5, l119: How did the authors decide RCP7.0 is closer to RCP8.5 than to RCP6.0? Also be consistent in using e.g. “RCP2.6” vs. “RCP 2.6”.

p5, l128ff.: Make clear you are using historic precipitation here.

p5, l129 and l140: I find the “(3.4)” and “(6.0)” confusing. I assume these numbers refer to the RCP but the RCP is already mentioned before. If you want to distinguish the two SSP4 scenarios better do it as for the SSPs and use something like “low-mid warming” and “mid-high warming”.

p5, l142 ff.: Why is global and not US population/demand/fertilizer relevant here? Is it that increasing food demand in the US is fulfilled in parts by imported food from other countries and this is only possible via low population growth/demand in other countries?

p7, l187ff: provide some numbers about the extent of bioenergy croplands, e.g. absolute area and the fraction of the total crop area. Do per-cropland-area fertilizer rates increase or decrease compared to scenarios without bioenergy (so is the fertilizer increase only because of cropland expansion)?

p7, l208: change to “land us as cropland” to “cropland area”.

p8, l217 ff./Fig. 4: I am quite surprised by the large impacts of cropland area in SSP5-8.5, a scenario I thought to be characterized by very limited cropland changes in the US. Why does this happen? Fig. S3 suggests that there are some common patterns in the LUH2 scenarios so is it just an effect of cropland abandonment between 2005 and 2015 (see also Fig. S4 and p7, l 208)? If so, can the decline really be attributed to “future” land use change while it actually already happened? Did agricultural abandonment occur in reality over this period?

p10, l303: “land use”

p13, l371: 2003-2006?

Table S4: Are the non-marker GCAM scenarios (or from other IAMs) also available for download?

Reviewer #3 (Remarks to the Author):

Sinha et al. examined the influence of changes in land use and management on the eutrophication in the 21st century, a major environment problem in the world. This is a satisfied research to date for such a large environment concern. Different from previous efforts, they used comprehensive scenario analyses and simulated the spatiotemporal variations of nitrogen (N) using the coupled models on a grid cell basis. They successfully evaluate the changes in the various N sources and export for the period 2020-2100. They also analyzed the increasing eutrophication in the world, including the risk in the Asia. The results clearly confirmed that human activities have substantially altered N delivery, cycling and export to ecosystems, which would cause a wide interest. The manuscript is well written, the results are well presented and the conclusions sound. I suggest that the manuscript can be accepted for publication in Nature Commutations.

A minor comment

Figure 3 could show the performance of the model, discrepancy among six scenarios. It seems a large uncertainties exist. Again, Asia may experience the great risk of eutrophication by the end of this century as shown in Figure 5. I can see the significance of this study. The study could be quite useful for policy makers to contend with the eutrophication. It, however, should be noted that the trends of fertilizer application are changing in Asia, especially in China. Be sure to explain whether such uncertainties influence the conclusion drawn from the modeled results.

We thank the reviewers for their insightful input, which has helped to further improve the manuscript. We have provided detailed responses to the reviewers' individual comments below. In all cases, line numbers refer to the revised version of the manuscript, even when referencing text that was already present in the original version.

Reviewers' comments:

Reviewer #1 (Remarks to the Author):

Comment 1.1: *The paper provided a global scale assessment of the combined effects of project climate change and land use on eutrophication. As far as I am aware, this is the first study to do so, and as such this makes a novel contribution to the literature which will be of interest to others in the community and the wider field of environmental sciences. The methodology used to analyse each scenario builds on earlier published work and is extremely well documented and detailed, is in my opinion valid and appropriate, and is rigorous in its approach. It treats well the uncertainty in climate change projection. The documentation, and the earlier published works, would enable another researcher to repeat the analysis.*

The conclusions are original, in that they find that climate mitigation may increase eutrophication, although some pathways allow for limiting both climate change and eutrophication. It is made clear that it is the increased use of biofuels that is the cause of the unintended impact of climate change mitigation. More generally, they also report that future societal choices will have a large impact on eutrophication, which is less surprising.

Response 1.1: We thank the reviewer for recognizing the importance and contribution of our work.

Comment 1.2: *The specific projections or a range of 1-68% increase in nitrogen loading, which result which climate change impacts on precipitation are also included, are quite startling. This statement should, however, be caveated since it is the range associated with the particular set of scenarios studied, and other scenarios can be envisaged that are not explored in the paper.*

Response 1.2: We agree with the reviewer that the range highly depends on the socioeconomic pathways selected and the IAM's used for combining these pathways with climate outcomes to produce a range of scenarios. This was described in lines 191-201 and 205-216 of the original manuscript. We have now also reiterated this in lines 166-167 of the revised manuscript.

Comment 1.3: *The paper explores in detail the effects of different levels of climate change mitigation (low, moderate, high), food production (low, moderate high, and generally associated with population trends), combined with climate change itself. Whilst it makes sense to consider climate change scenarios which match the level of climate change mitigation being modelled, some of the combinations of mitigation and food production are not explored. For example, there is no high mitigation and low food demand scenario; and there is no low mitigation and high food production scenario. The two extremes appear to be high mitigation and moderate food production; and low mitigation with no increase in food production. In particular, with regard to the SSP5-8.5 scenario, it seems that the assumption that is made, explained on p 12, that fertilizer use decreases due to an increase in productivity, seems to be a strong driver of the results - ie.this result is an inevitable consequence of the input assumption which seems optimistic. I would argue that in SSP5, where there is high food demand, it would be more likely*

that fertiliser use would increase. Indeed, this assumption seems counter-intuitive and it is the output of this scenario that suggests that lack of climate change mitigation would reduce eutrophication. Hence it would be useful to explore the combination of RCP8.5 with other assumptions about population and food demand. SSP5 is not the only SSP that is generally considered compatible with RCP8.5.

Response 1.3: The SSP1-2.6 considered in our analysis represents a high mitigation and low food demand scenario while the SSP3-7.0 represents a low mitigation and high food demand scenario. Table R1 below summarizes (modified from Table 1, 2, and 3 in O'Neill et al., 2017, which is also referenced in lines 87, 352, and 561 of the manuscript) some of the assumptions about population, food consumption, environmental policy, fossil fuel dependence, land use change, and agriculture productivity for the five socioeconomic pathways.

We agree with the reviewer that if the SSP5 socioeconomic pathway with high fossil fuel dependence, and thus presenting a large challenge for mitigation, had high food demand, this would likely result in higher fertilizer usage. However, the SSP3 pathway already explores this combination as this pathway has high dependence on fossil fuel and thus has low mitigation along with high food demand that is driven by material-intensive consumption. This was described in lines 173-176 of the original manuscript. We have further edited this line to make it clearer.

We also agree with the reviewer that SSP5 pathway is not the only pathway that is considered compatible with the RCP8.5 climate outcome. However, the Tier I and Tier II scenarios generated for utilization in CMIP6 have only developed one scenario that combines an SSP pathway with the RCP8.5 climate outcome, namely the SSP5-8.5 scenario (Figure 2 in O'Neill et al., 2016).

Table R1: Summary of assumptions used in the development of socioeconomic pathways. Based on summary tables provided in O'Neill et al., 2017. OECD stands for Organization for Economic Co-operation and Development; LIC stands for low-income countries; MIC for medium-income countries; and HIC for high-income countries.

SSP element	SSP1	SSP2	SSP3	SSP4	SSP5
Population growth	• Relatively low	• Medium	• High for all except rich-OECD	• Relatively high for all except rich-OECD	• Relatively low
Consumption & Diet	• Low growth in material consumption • Low-meat diet	• Material-intensive consumption • Medium meat consumption	• Material-intensive consumption	• High consumption by elites but low consumption by rest	• High material consumption. • Meat-rich diets
Environmental Policy	• Tighter regulations aimed at improving environmental conditions	• Moderate success in improving environmental conditions.	• Low priority for environmental issues	• Environmental policies focus only on local issues in MICs and HICs.	• Environmental policies focus only on local issues
Carbon intensity	• Low	• Medium	• High in regions with large fossil fuel resources	• Low/medium	• High
Land Use	• Strong regulations	• Medium regulations; • Deforestation decreases at a slow rate	• Hardly any regulation • Continued deforestation	• High regulation in MICs and HICs • No regulation in LICs	• Medium regulations; • Deforestation decreases at a slow rate
Agriculture	• Improved productivity • Best practices implemented	• Technological advances at a medium pace	• Low technology development	• High productivity for large-scale farming but low for small-scale farming.	• Rapid increase in productivity

- O'Neill, B. C. *et al.* The scenario model intercomparison project (scenariomip) for CMIP6. *Geosci Model Dev* **9**, 3461 (2016).
- O'Neill, B. C. *et al.* The roads ahead: Narratives for shared socioeconomic pathways describing world futures in the 21st century. *Global Environ. Change* **42**, 169–180 (2017).

Comment 1.4: *Similarly, the issue of the extent to which climate change mitigation could be achieved without a significant increase in biofuel use, or not, needs to be explored and discussed, and brought out much more. Clearly the authors only had access to SSP1-2.6, but a wide range of other scenarios which achieve stringent mitigation targets have emerged in the past year or so, and whilst I am not suggesting that the authors should simulate these, they should discuss this literature in particular focusing on the amount of biofuel use. A particularly relevant paper here is Grubler A, Wilson C, Bento N, Boza-Kiss B, Krey V, McCollum D, Rao ND, Riahi K, et al (2018). "A Low Energy Demand Scenario for Meeting the 1.5°C Target and Sustainable Development Goals without Negative Emission Technologies." Nature Energy. I think it would be good to bring out more a message that climate change mitigation would need to be done in certain ways in order to avoid unintended side effects. Minimising the use of biofuel would also avoid many of the other undesirable side effects - when used at large scales, competition with food production and biodiversity conservation would occur. Hence there is actually a win win option out there which avoids all of these unintended consequences. At the moment the message of the paper is a little misleading, in that it would be easy to take away a message that policy makers need to choose between climate change and eutrophication, or to balance the two. At the moment the results show that some mitigation scenarios increase eutrophication and that some non-mitigation scenarios decrease it, but it may well be that the reverse is also true.*

Response 1.4: Two of the scenarios considered in our analysis are based on high mitigation pathways: SSP1-2.6 and SSP4-3.4. For the continental United States the SSP4-3.4 achieves climate mitigation goals through increased biofuel production, which results in an increase in TN loading by 54%. The SSP1-26, however, has lower biofuel expansion.

We agree with the reviewer that it is indeed possible to achieve climate mitigation while reducing the harmful impacts on the environment and that this message needs to come out more clearly in our text. We have added the reference provided by the reviewer and added additional details in lines 168-170 and 178-179 to better convey this message.

We also agree with the reviewer's last comment that, "*At the moment the results show that some mitigation scenarios increase eutrophication and some non-mitigation scenarios decrease it, but it may well be that the reverse is also true*". This is indeed shown in our results for the SSP4-6.0 and SSP3-7.0 scenarios that result in substantial potential increases in eutrophication along with low climate mitigation.

Comment 1.5: *So, the analysis of the LUH2 marker scenarios needs to be complemented by a discussion of the limitations of the scenario combinations explored, which means that more outcomes are possible than those reported upon, and the authors could speculate on what these might be. The setting of cropland area which are impose, are strong determinants of the results, for example. How robust are the results of these assumptions and what is the rationale behind them? This needs to be explained in more detail.*

Response 1.5: We agree that our results are dependent on the range of societal outcomes considered in the SSP scenarios and additional scenarios including the non-marker scenarios and scenarios that combine the five SSP pathways with different climate outcomes may exist that are outside of the considered range (e.g., Popp et al., 2017; Riahi et al., 2017). We have added this caveat in lines 166-167 of the paragraph that addresses uncertainty added by the IAM used to implement the SSP.

- Popp, A. *et al.* Land-use futures in the shared socio-economic pathways. *Global Environ. Change* **42**, 331–345 (2017).
- Riahi, K. *et al.* The Shared Socioeconomic Pathways and their energy, land use, and greenhouse gas emissions implications: An overview. *Global Environ. Change* **42**, 153–168 (2017).

Comment 1.6: *In Table S2, a sensitivity analysis is provided which contains fertiliser usage projected by GCAM. These scenarios are, as the authors state, very valuable, but they need to be taken through the full analysis (as done for the LUH2 scenarios) in order to explore the full implications of this uncertainty.*

Some aspects of uncertainty (such as uncertainty across CMIP5 models) are well explored, and it would be a pity not to complete the picture in this way.

Response 1.6: We have not converted the GCAM outputs to TN flux estimates because that will not alter the main message, which is that the outputs (fertilizer usage, land use change, or TN fluxes) are highly dependent on the IAM used to represent the pathway. The GCAM model implementation of the SSP1-2.6 and SSP2-4.5 scenarios result in much larger cropland area and fertilizer usage relative to the corresponding marker scenarios in the LUH2 database (Figure 3 in the manuscript), which will translate into much higher TN flux estimates for the GCAM based scenarios than the marker scenarios in the LUH2 database.

Furthermore, the LUH2 has only released harmonized set of outputs for a single marker scenario for each SSP and thus, harmonized GCAM outputs for all six scenarios are not available, precluding a direct comparison of loading. We agree that the reason for not estimating TN flux estimates using the GCAM outputs had not been clearly described in the manuscript, however, and we have now added this reasoning in lines 201-205.

Comment 1.7: *Finally, the discussion would benefit from considering existing legislation about eutrophication, especially in relation to the Figure 5 for Asia. Are there particular pieces of legislation which would be violated, eg in the US or the EU?*

Response 1.7: While the policy and legal setting of future nitrogen loading is highly interesting, it is outside the primary focus of this manuscript. At a high level, both China and India are primarily focused on improving nutrition and ensuring food security for its population, and no national or state level legislation for reducing nitrogen inputs exists for China and India. However, in recent years, there have been efforts by the Chinese government to reduce nitrogen pollution by engaging small farm holders to adopt management practices that increase crop yield while reducing adverse environmental impacts (e.g., Zhang et al., 2016, Cui et al., 2018). In India, the central government has launched a scheme that promotes organic farming (*Paramparagat Krishi Vikas Yojana* <https://darpg.gov.in/sites/default/files/Paramparagat%20Krishi%20Vikas%20Yojana.pdf>) that is aimed at producing chemical free products.

Our analysis shows that for most of the SSP scenarios both these nations will likely experience large increase in fertilizer usage (Figure 5 in the manuscript). This finding implies that efforts should be focused on finding sustainable ways of feeding the large population that limits adverse environmental impacts. We have added this implication in lines 310-312, and also added the references cited above in lines 294-299 to clarify again that the SSPs used here do not cover the

full range of possible outcomes, which will depend in no small part on policy and legislative action by individual countries.

- Cui, Z. et al. Pursuing sustainable productivity with millions of smallholder farmers. *Nature* 555, 363–366 (2018)
- Zhang, W. et al. Closing yield gaps in China by empowering smallholder farmers. *Nature* 537, 671–674 (2016)

Minor comments:

Comment 1.8: *In the results section, at the beginning it is not immediately clear that the effect of changes in climate are not driving the results at this point. As one reads on, it becomes clear that this part of the discussion relates only to the land use and climate mitigation policy drivers, and that climate change effects are added later. I suggest splitting the results section clearly into two to differentiate between these two sections.*

Response 1.8: We have divided the results into two sections and also made it clear that in the first section the impacts of changes in precipitation patterns are not considered (line 138).

Comment 1.9: *My overall opinion is that with some more calculations and appropriate modification to the discussion and conclusions to incorporate the very important points above, that the paper is suitable for publication in Nature Communications. This is necessary to ensure that the policy relevant messages are truly objective and not skewed by the scenarios that happen to be readily available for analysis. This will be important because this paper will undoubtedly influence thinking in the field.*

Response 1.9: We thank the reviewer for their constructive input for improving the clarity of the manuscript, and for agreeing that our paper is suitable for publication in *Nature Communications*. Details of how we addressed various comments are provided in the responses above.

Reviewer #2 (Remarks to the Author):

Comment 2.1: *Sinha et al., based on an updated empirical model used in previous studies, investigate the effects of future land use and land management on nitrogen loading. They found that, depending on the selected LUH2 land use scenario, end-of-century N loading in the US will increase by up to 54%, including large regional differences. In addition, the authors study the combined effects of changes in land use and precipitation and investigate the isolated effects of changes in cropland area, urban area, and fertilizer rates. Sinha et al. also provide some valuable insights about the substantial differences across integrated assessment models when simulating the same SSP-RCP scenario. The study is well-written and the results highlight the relevance of future land use changes on riverine nitrogen loading, even though at some points I wondered how trustworthy the results are given the limited input data. I find the study worth for publication after some clarifications.*

Response 2.1: We thank the reviewer for their positive feedback on our work. We have revised our manuscript as per the reviewer's comments.

Comment 2.2: *One aspect I find somewhat worrying is that by applying the very wide range of land use projections from IAMs/LUH2, substantial changes and differences across scenarios for all kinds of environmental variables are naturally to be expected, but we actually have no clue*

which of the projections are realistic. I understand that the scope of this study is not to investigate the plausibility of the scenarios in detail but could the authors emphasize some of the underlying IAM assumptions (apart from “demand for bioenergy drives cropland and fertilizer increase...”, e.g. the assumed rate of yield improvements, nitrogen use efficiency, food demand in the US, what would be implications of large-scale bioenergy cultivation in SSP4-RCP3.4 on food production?) and why there is such a large discrepancy before and after harmonization (I would not call a difference of 14.6 vs. 11.5 TgN yr⁻¹ in year 2015 “small”). How do the projections fit to what happened in the past few years?

Response 2.2: Various underlying IAM assumptions are discussed in detail in a number of papers published in the special issue of *Global Environment Change* (42) 2017. For example:

- Kriegler et al., 2017 summarizes the assumption behind the implementation of the SSP5 marker scenario. The projected increase in per-capita food demand and consumption for North America is shown in Supplementary Online Material (SOM) Figure S3.6 of this manuscript. While the livestock productivity increase and the soil nitrogen uptake efficiency improvement in North America are shown in SOM Fig S3.7 and Fig S3.8, respectively.
- Calvin et al., 2017 summarizes the assumption behind the implementation of the SSP4 marker scenarios. The increase in cropland area and fertilizer usage in the high-income regions (HIR) resulting from the increase in food demand is summarized in Table 1, Section 3.2, and SOM Fig S3 of Calvin et al., 2017. The increase in biofuel energy production and associated increase in cropland area are shown in SOM Fig 14 and 15 for the mitigation scenario SSP4-3.4.
- Popp et al., 2017 summarizes land-related results for all IAMs. Differences in models implementation of the SSPs are articulated in a supplementary spreadsheet. Additionally, SOM Table SI.1 and SI.2 summarize differences across SSPs.

In the revised manuscript, we have further emphasized in lines 351-354 that the IAM estimates are based on a number of assumptions and cited the papers listed above. However, a full description of IAM assumptions is beyond the scope of the current work. Similarly, a full comparison of IAM results to recent history is beyond the scope of this paper. Instead, we refer the reader to the original SSP papers (e.g., Kriegler et al., 2017; Calvin et al., 2017), which show the future results together with historical trends.

The historical trajectory of the scaled LUH2-based fertilizer usage (source: Zhang et al., 2015) matches closely with the USGS estimates (Figure R1; see also Figure S5 in the original manuscript). The LUH2 based fertilizer usage was scaled such that estimates from 1987-2012 period (obtained from LUH2 v2h) matched USGS estimates of fertilizer usage for each HUC2 region, as described in Section “Future changes in TN flux due to change in land use and land management” of the original manuscript. GCAM, however, uses information from the International Fertilizer Industry Association (Heffer et al., 2009).

The need for harmonized land use database is mentioned in Lawrence et al., 2017. The manuscript summarizing the LUH2 harmonization process is still in preparation (Hurtt et al., In prep.) and will likely explain the cause of large discrepancy before and after harmonization.

- Kriegler, E. *et al.* Fossil-fueled development (SSP5): An energy and resource intensive scenario for the 21st century. *Global Environ. Change* **42**, 297–315 (2017).
- Calvin, K. *et al.* The SSP4: A world of deepening inequality. *Global Environ. Change* **42**, 284–296 (2017).

- Popp, A. *et al.* Land-use futures in the shared socio-economic pathways. *Global Environ. Change* **42**, 331–345 (2017).
- Zhang, X. *et al.* Managing nitrogen for sustainable development. *Nature* **528**, 51–59 (2015).
- Heffer, P. *Assessment of Fertilizer Use by Crop at the Global Level 2009/07-2008/08*. (International Fertilizer Industry Association, 2009).
- Lawrence, D. M. *et al.* The Land Use Model Intercomparison Project (LUMIP) contribution to CMIP6: rationale and experimental design. *Geosci. Model Dev.* **9**, 2973–2998 (2016).
- G. Hurtt, L. Chini, R. Sahajpal, S. Frolking, et al. Harmonization of global land-use change and management for the period 850-2100. Geoscientific Model Development (In prep).

Figure R1: Time series of fertilizer usage for the historical (light blue background) period based on USGS estimates (brown, see section “Historical and future NANI datasets”) and the LUH2 database (black, see section “Future changes in TN flux due to change in land use and land management”), as well as for the middle-of-the-century (light red background) and end-of-the-century (light green background) time periods for the six scenarios considered here. Time series are shown for the continental United States (CONUS).

Comment 2.3: *In addition, not being familiar with the model, I find parts of the method section quite difficult to follow. For instance, I do not understand the component “net food and feed import”, can you explain it briefly? Is it the nitrogen taken up by plants and then removed from the system via harvest/grazing (and eventually returned elsewhere –if so where?)? Same for “non food crop export”, why is it assumed to be constant? Shouldn’t this vary (and not be extremely small) with the extent of bioenergy crops? Or is bioenergy part of food and feed? Furthermore, the authors simulate future food and feed import/export as a function of future N fertilization/fixation and the relationship seems to be ok for present-day conditions but will this also hold for the future under changing environmental conditions and crop types? Do the IAMs assume increased nitrogen use efficiency and/or different fertilizer rates for bioenergy crops and food crops and how will this affect simulated food and feed import?*

Response 2.3: Net food and feed import accounts for crop and animal N production (removal of N from the watershed and therefore negative flux) and human and animal N consumption (addition of N to the watershed and therefore positive flux) while non-food crop export represents the nitrogen in cotton and tobacco harvested for sale and exported out of the watershed area for non-food use. The non-food crop export represents nitrogen leaving the watershed and its value is subtracted from NANI while all other components are added to estimate NANI. We agree that non-food crop export will vary with time but historically it has been a very small portion of NANI (on average 0.78% of NANI). We have added definitions of net food and feed import and non-food crop export on lines 383-387 and line 415 of the revised manuscript. The breakdown of all NANI components on average during the historical period is as follows: fertilizer is 72% of NANI; fixation is 45% of NANI; deposition is 14% of NANI; food and feed import is -30% of NANI (note negative sign designating net export over the continental United States); and non-food crop export is 0.78% of NANI. The contribution of individual components of NANI to the total future NANI is also shown in Figure S6 of the manuscript. We have added the breakdown of various NANI components on lines 387-390 of the revised manuscript.

Bioenergy production and consumption is accounted for as part of net food and feed import, and we have clarified this in the revised manuscript in lines 385 and 497. We also agree that future trade policies will impact where bioenergy is consumed but because how trade policies will change in the future cannot be predicted, here we have estimated future food and feed import under the assumption that the relationship between food and feed import and fertilizer and nitrogen fixation will remain constant over time, because no better sources and estimates are available at this time. This stationarity assumption is similar to the ones used in climate models, and is described in Section “Future changes in TN flux due to change in land use and land management” and lines 498-500 of the manuscript.

The LUH2 database provides cropland area and fertilizer application rate for five different crop types. Total crop production (whether for biofuels or for food and feed) is broken out according to these five categories. Biofuel crops and food crops are each broken out into these five categories. The cropland area and fertilizer application rate changes over time for the five different crop types (see Figure R2 and R3). Fertilizer application rates that decrease over time represents improvement in nitrogen use efficiency (e.g., C3 and C4 annual crops for SSP5-8.5 scenario). Because we have estimated food and feed import as a function of fertilizer usage and nitrogen fixation, reduction in fertilizer usage resulting from either improvement in nitrogen use

efficiency or reduction in total cropland area will also impact the nitrogen embedded in net food and feed import. This has been clarified in lines 504-507 of the revised manuscript.

Figure R2: Time series of fertilizer application rate for the five different crop types based on the LUH2 database for the continental United States for the historical (light blue background) period, the middle-of-the-century (light red background), and end-of-the-century (light green background) time period for the six scenarios examined here.

Figure R3: Time series of cropland area for the five different crop types based on the LUH2 database for the continental United States for the historical (light blue background) period, the middle-of-the-century (light red background), and end-of-the-century (light green background) time period for the six scenarios examined here.

Minor comments:

Comment 2.4: *p5, l119: How did the authors decide RCP7.0 is closer to RCP8.5 than to RCP6.0?*

Response 2.4: SSP3 combines "high societal vulnerability with high forcing" (O'Neill et al., 2016) and therefore we paired SSP3-7.0 with RCP8.5 with high forcing rather than RCP6.0 that has medium forcing. We have added this clarification in lines 124-126.

- O'Neill, B. C. *et al.* The scenario model intercomparison project (scenariomip) for CMIP6. *Geosci Model Dev* **9**, 3461 (2016).

Comment 2.5: *Also be consistent in using e.g. "RCP2.6" vs. "RCP 2.6"*

Response 2.5: We have made this consistent throughout the manuscript.

Comment 2.6: *p5, l128ff.: Make clear you are using historic precipitation here.*

Response 2.6: As suggested by Reviewer #1 (Comment 1.8) we have divided the results into two sections and also made it clear that in the first section the impacts of change in precipitation patterns are not considered (line 138).

Comment 2.7: *p5, l129 and l140: I find the "(3.4)" and "(6.0)" confusing. I assume these numbers refer to the RCP but the RCP is already mentioned before. If you want to distinguish the two SSP4 scenarios better do it as for the SSPs and use something like "low-mid warming" and "mid-high warming".*

Response 2.7: The 3.4 in SSP4-3.4 and 6.0 in SSP4-6.0 refers to the corresponding climate outcome, i.e., these scenarios will result in net radiative forcing of 3.4 and 6.0 W m⁻² in 2100. We have followed the nomenclature used by O'Neill et al., 2016. This clarification has been added on lines 105-107 of the revised manuscript.

- O'Neill, B. C. *et al.* The scenario model intercomparison project (scenariomip) for CMIP6. *Geosci Model Dev* **9**, 3461 (2016).

Comment 2.8: *p5, l142 ff.: Why is global and not US population/demand/fertilizer relevant here? Is it that increasing food demand in the US is fulfilled in parts by imported food from other countries and this is only possible via low population growth/demand in other countries?*

Response 2.8: In SSP5 population growth is assumed to be low for the United States as well as for other parts of the globe. We have edited this sentence in the manuscript accordingly.

Comment 2.9: *p7, l187ff: provide some numbers about the extent of bioenergy croplands, e.g. absolute area and the fraction of the total crop area. Do per-cropland-area fertilizer rates increase or decrease compared to scenarios without bioenergy (so is the fertilizer increase only because of cropland expansion)?*

Response 2.9: Time series of 2nd generation biofuels area and food crop as a percentage of the total cropland area and in square km is shown in Figure R4 and R5, respectively. Fertilizer usage per unit of cropland area increases for SSP scenarios both with and without bioenergy crops. For example, SSP4-3.4 is projected to experience large increase in 2nd generation biofuels (Figure R4 and R5) that falls under the C4 perennial crops (Fig R3) along with a slightly increasing fertilizer application rate for the C4 perennial crops for the future time period (Figure R2). On the other hand, the SSP2-4.5 and SSP3-7.0 are not projected to see an increase in 2nd generation biofuels

(Figure R4 and R5) but are projected to see an increase in fertilizer application rate for the C4 perennial (SSP2-4.5), C3 nitrogen-fixing (SSP2-4.5), and C3 annual (SSP3-7.0) crops (Figure R2). For SSP5-8.5, with no projected increase in 2nd generation biofuels, the cropland area remains unchanged for various crop types (Figure R3); however, the fertilizer application rates decreases for the majority of the crop types signifying improvement in fertilizer use efficiency (Figure R2). We have added the 2nd generation biofuel crop extent Figure R5 in our revised manuscript as Figure S8 and referenced it in line 242.

Figure R4: Time series of percentage of the cropland area used for 2nd generation biofuels based on the LUH2 database for the continental United States for the middle-of-the-century (light red background) and end-of-the-century (light green background) time period for the six scenarios examined here.

Figure R5: Time series of land area used for 2nd generation biofuels based on the LUH2 database for the continental United States for the middle-of-the-century (light red background) and end-of-the-century (light green background) time period for the six scenarios examined here.

Comment 2.10: p7, l208: change to “land us as cropland” to “cropland area”.

Response 2.10: Done.

Comment 2.11: p8, l217 ff./Fig. 4: I am quite surprised by the large impacts of cropland area in SSP5-8.5, a scenario I thought to be characterized by very limited cropland changes in the US. Why does this happen? Fig. S3 suggests that there are some common patterns in the LUH2 scenarios so is it just an effect of cropland abandonment between 2005 and 2015 (see also Fig. S4 and p7, l 208)? If so, can the decline really be attributed to “future” land use change while it actually already happened? Did agricultural abandonment occur in reality over this period?

Response 2.11: The reviewer is correct, that the RCP8.5 scenario projects minimal change in cropland area. The decrease in the future MARB loading is driven by the large reduction in cropland area from 1997 to 2015 (Figure S4). Because of this reduction, the average cropland area over the historical period is larger than the projected cropland area, which results in a decrease in TN loading by the end of the century as compared to the historical period. The cause of reduction in MARB loading in the future time periods attributable to reduction in cropland

area in the historical period was explained in lines 224-227 of the original manuscript but the cropland reduction period was too narrowly stated as 2005-2015. We have corrected this range to 1997-2015 (line 226 of the revised manuscript).

The decrease in cropland area from 1997 to 2012 is well documented in the USDA reports (e.g., Nickerson et al., 2011; Bigelow and Borchers, 2017). According to these reports the cropland area dropped by 13 million acres from 1997 to 2002, by 34 million acres from 2002 to 2007, and by 16 million from 2007 to 2012.

- Nickerson, C., Ebel, R., Borchers, A. & Carriazo, F. *Major uses of land in the United States, 2007*. (USDA Economic Research Service, 2011).
- Bigelow, D. P. & Borchers, A. *Major Uses of Land in the United States, 2012*. (U.S., Department of Agriculture, Economic Research Service, 2017).

Comment 2.12: p10, l303: “land use”

Response 2.12: Done.

Comment 2.13: p13, l371: 2003-2006?

Response 2.13: We appreciate that the reviewer caught this minor typo and have fixed it.

Comment 2.14: Table S4: Are the non-marker GCAM scenarios (or from other IAMs) also available for download?

Response 2.14: We obtained the outputs for the non-marker GCAM scenario from the lead modeler and scenario developer of the GCAM model, Dr. Kate Calvin, who is also a co-author of this manuscript. These will be freely available to anyone who wishes to use them upon publication of this manuscript. The outputs for the non-marker scenarios are not harmonized by the LUH2 team and are thus not available to download from their website (<http://luh.umd.edu/index.shtml>).

Reviewer #3 (Remarks to the Author):

Comment 3.1: *Sinha et al. examined the influence of changes in land use and management on the eutrophication in the 21st century, a major environment problem in the world. This is a satisfied research to date for such a large environment concern. Different from previous efforts, they used comprehensive scenario analyses and simulated the spatiotemporal variations of nitrogen (N) using the coupled models on a grid cell basis. They successfully evaluate the changes in the various N sources and export for the period 2020-2100. They also analyzed the increasing eutrophication in the world, including the risk in the Asia. The results clearly confirmed that human activities have substantially altered N delivery, cycling and export to ecosystems, which would cause a wide interest. The manuscript is well written, the results are well presented and the conclusions sound. I suggest that the manuscript can be accepted for publication in Nature Communications.*

Response 3.1: We thank the reviewer for recognizing the novelty, importance, and contributions of this work.

A minor comment

Comment 3.2: *Figure 3 could show the performance of the model, discrepancy among six scenarios. It seems a large uncertainties exist. Again, Asia may experience the great risk of eutrophication by the end of this century as shown in Figure 5. I can see the significance of this study. The study could be quite useful for policy makers to contend with the eutrophication. It, however, should be noted that the trends of fertilizer application are changing in Asia, especially in China. Be sure to explain whether such uncertainties influence the conclusion drawn from the modeled results.*

Response 3.2: The reviewer has correctly interpreted the key results presented in Figures 3 and 5, and we thank the reviewer for recognizing both the scientific and policy relevance of the work. With regard to Figure 3, if the reviewer was also asking about why the GCAM output is presented in terms of fertilizer usage rather than nitrogen loading, then please refer to the response to Comment 1.6 by Reviewer #1.

We agree with the reviewer that fertilizer usage trends have been changing in Asia, with fertilizer usage in China increasing from 11.2 Tg yr⁻¹ in 1980 to 32.1 Tg yr⁻¹ in 2010 (Gu et al., 2015) and in India increasing from 1.5 Mg yr⁻¹ in 1970 to 17.4 Tg yr⁻¹ in 2015 (Ministry of Agriculture & Farmers Welfare, 2017). The fertilizer usage is projected to further increase with an annual growth rate for nitrogen fertilizer usage expected to be 0.9% for East Asia and 2.5% for South Asia from 2015-2020 (FAO, 2017).

However, large uncertainties do exist in projected fertilizer usage trends for China and India because of two primary reasons. First, the projected fertilizer usage is highly depended on the range of socioeconomic pathways selected, the IAM used for implementing them, and the climate outcomes with which the pathways are combined to produce scenarios. These uncertainties are explained in more detail in Response 1.2 and 1.5 above. Second, future nitrogen use efficiency may improve in both China and India as both countries are now increasing efforts on reducing nitrogen pollution as explained in Response 1.7 above.

- Gu, B., Ju, X., Chang, J., Ge, Y. & Vitousek, P. M. Integrated reactive nitrogen budgets and future trends in China. *Proc. Natl. Acad. Sci. U.S.A.* **112**, 8792–8797 (2015).
- Ministry of Agriculture & Farmers Welfare. *Agricultural statistics at a glance 2016*. 519 (Government of India, 2017).
- FAO. *World fertilizer trends and outlook to 2020: Summary Report*. (FAO, 2017).

REVIEWERS' COMMENTS:

Reviewer #2 (Remarks to the Author):

I am satisfied with the authors resubmission.

Reviewer #3 (Remarks to the Author):

The revised manuscript is well written, the results are well presented and the conclusions sound. And all the comments are well addressed. I am pleased to suggest that the manuscript can be accepted for publication as it is.